Novel cross-dimensional coarse-fine-grained complementary network for image-text matching

Liu Meizhen 1 2
http://orcid.org/0000-0002-9873-4779 Khairuddin Anis Salwa Mohd 1 anissalwa@um.edu.my
Hasikin Khairunnisa 3
Liu Weitong 1 2
1 Department of Electrical Engineering, Faculty of Engineering, Universiti Malaya , Kuala Lumpur , Malaysia
2 School of Data and Computer Science, Shandong Women’s University , Jinan, Shandong , China
3 Department of Biomedical Engineering, Faculty of Engineering, Universiti Malaya , Kuala Lumpur , Malaysia
Somani Arun
Electronic publication date: 2025 Mar 3
Publication date: 2025
Volume: 11
Electronic Location ID: e2725
Received 2024 Jul 9; Accepted 2025 Jan 31
Copyright: © 2025 Liu et al.
Copyright year: 2025
Copyright holder: Liu et al.
License: This is an open access article distributed under the terms of the Creative Commons Attribution License, which permits unrestricted use, distribution, reproduction and adaptation in any medium and for any purpose provided that it is properly attributed. For attribution, the original author(s), title, publication source (PeerJ Computer Science) and either DOI or URL of the article must be cited.
License URL: https://creativecommons.org/licenses/by/4.0/

Keywords: Complementary, Image-text matching, Cross-dimensional, Semantic aggregation, Semantic consistency

Funding: The authors received no funding for this work.

==============================
The fundamental aspects of multimodal applications such as image-text matching, and cross-modal heterogeneity gap between images and texts have always been challenging and complex. Researchers strive to overcome the challenges by proposing numerous significant efforts directed toward narrowing the semantic gap between visual and textual modalities. However, existing methods are usually limited to computing the similarity between images (image regions) and text (text words), ignoring the semantic consistency between fine-grained matching of word regions and coarse-grained overall matching of image and text. Additionally, these methods often ignore the semantic differences across different feature dimensions. Such limitations may result in an overemphasis on specific details at the expense of holistic understanding during image-text matching. To tackle this challenge, this article proposes a new Cross-Dimensional Coarse-Fine-Grained Complementary Network (CDGCN). Firstly, the proposed CDGCN performs fine-grained semantic alignment of image regions and sentence words based on cross-dimensional dependencies. Next, a Coarse-Grained Cross-Dimensional Semantic Aggregation module (CGDSA) is developed to complement local alignment with global image-text matching ensuring semantic consistency. This module aggregates local features across different dimensions as well as within the same dimension to form coherent global features, thus preserving the semantic integrity of the information. The proposed CDGCN is evaluated on two multimodal datasets, Flickr30K and MS-COCO against state-of-the-art methods. The proposed CDGCN achieved substantial improvements with performance increment of 7.7–16% for both datasets.

Introduction

As the volume of digital information increases rapidly, research on images and texts has become a prominent focus within the multimodal applications (Yadav & Vishwakarma, 2024; Zhang et al., 2024a), attracting more scholars to work on cross-modal studies involving vision and language. This includes key areas such as cross-modal retrieval (Hu et al., 2023; Ma et al., 2024; Sun et al., 2023; Zeng et al., 2022), image captioning (Li et al., 2024a; Ma et al., 2023; Yan et al., 2021; Yu et al., 2019) and visual question answering (Dong et al., 2021; Li et al., 2022b; Liu, Li & Lin, 2023; Qian et al., 2024) multimodal tasks. These works focused on the core challenge of matching images and text, which is a fundamental task in cross-modal retrieval. It involves finding the most pertinent sentences (or images) in a dataset based on a query image (or sentence). The image-text matching task has found widespread application in various fields, including information retrieval, social media analysis, and intelligent recommendation systems (Chhabra & Vishwakarma, 2023; Hu & Huang, 2021; Moro, Salvatori & Frisoni, 2023).

In recent times, the image-text matching task has remained a focal point of extensive research, owing to the considerable heterogeneity gap between visual and textual data. Despite proposing numerous attempts to tackle this problem with various methods (Jeong et al., 2024; Wang et al., 2023a; Wu et al., 2024; Yao et al., 2024; Zhao et al., 2023), several challenges still persist. Firstly, existing methods often rely too heavily on a single type of pooling method, failing to effectively explore and utilize modal dimensional information, which lead to the loss of some essential discriminative information. Secondly, in pursuit of high accuracy, existing methods overly focus on capturing detailed information in images and texts, resulting in misalignment between global and local information matching of the image and text.

To address these challenges, our work introduces a cross-dimensional coarse-fine-grained complementary network for image-text matching. This approach provides a novel perspective on image-text alignment and serves as a valuable reference for other pertinent research endeavors in the realm of multimodal learning (such as video-text matching and audio-text matching). Additionally, the potential real-world applications of this work encompasses product recommendations, image search engines and social media content analysis. Thus, by improving the accuracy of information retrieval, this method allows users to access relevant content with improved accuracy and efficiency.

Specifically, existing methods can generally be categorized into two main groups: coarse-grained global matching and fine-grained local matching. The coarse-grained global method indicated in Fig. 1A aligns the entire image-text pair by mapping the visual and textual data to a shared space, unifying the semantic representation of the image and sentence. The advantages of coarse-grained global matching methods such as average pooling (Li et al., 2019), maximum pooling, and Generalized Pooling Operator (GPO) (Chen et al., 2021) are flexibility, simplicity, and short runtime. However, those pooling methods ignore that different dimensions of a feature could carry distinct interpretations, and the significance of various dimensions with the same feature might differ, hence, may further reduce discriminative information. For instance, Zhu et al. (2023) pointed out in the external spatial attention aggregation (ESA) model that existing pooling methods often use vector weighting to obtain global information from local features, unable to focus on dimensions at a fine-grained level, which may discard crucial detail information, resulting in poor retrieval performance. Wu et al. (2023) discovered that different dimensions of a feature may have different meanings, and the importance of different dimensions within the same feature should vary when aggregating features. Zhang et al. (2023) proved that dimensions are not independent and better feature extraction can be achieved by jointly representing latent semantics through intrinsic dependencies between dimensions. These studies suggest that existing coarse-grained pooling methods may unintentionally lose important discriminative information. Meanwhile, the fine-grained local method depicted in Fig. 1B emphasizes aligning relevant image-text pairs by utilizing the correspondence between local information (i.e., visual regions and text words). This method has high matching accuracy and is currently the most popular image-text matching method. Still, if the fine-grained local matching method is used alone, it will lead to excessive attention to the relationship between regions and texts while ignoring the semantic consistency problem between global and local information.

Figure 1 The general frameworks of image text matching.

(A) The coarse-grained global method by aligning the entire image text pair, (B) the fine-grained local method by aligning relevant image text pairs.

To address the shortcomings of relying solely on either global matching or local matching methods, mining richer cross-dimensional features from images and texts, this work proposes the CDGCN method as shown in Fig. 2, which uses fine-grained cross-dimensional semantic dependency branch to obtain the local similarity of region-words and then semantic matching, and coarse-grained cross-dimensional semantic aggregation branch to receive the global semantic matching of images and texts, the global semantic matching is used as a supplement to the local semantic alignment to achieve global and local semantic information consistency. This approach distinguishes information elements at the fine-grained similarity dimension level and aggregates features at the coarse-grained dimension level, thus increasing the accuracy of cross-modal retrieval and significantly improving the application potential of multimodal learning.

Figure 2 The proposed CDGCN logic framework of image-text matching.

The primary contributions of this article are outlined as follows: (1) A Cross-Dimensional Coarse-Fine-Grained complementary network (CDGCN) is proposed, integrating coarse-grained cross-dimensional global semantic matching across dimensions with fine-grained local region word cross-dimensional semantic alignment.

(2) A plug-and-play Coarse-Grained Cross-Dimensional Semantic Aggregation Strategy (CGDSA) is introduced. This strategy aggregates local features into global features by employing cross-dimensional vector aggregation weights across different dimensions and similar dimension.

(3) Extensive experiments and comparisons were performed on cross-validated datasets including Flick30k and MSCOCO, to demonstrate the advantages of the proposed CDGCN in cross-modal image-text matching.

The structure of the paper is as follows: “Related Work” provides an overview of pertinent research in cross-modal image-text retrieval. “Methodology” elaborates on the CDGCN method, detailing its components and functionality. “Results and Analysis” presents experimental findings, encompassing comparative analyses, ablation studies, hyper-parameter investigations, visualization and analysis. Finally, “Conclusion” offers a comprehensive summary and future scope of our work.

Related work

Regarding the progressive work on image-text matching and the extraordinary performance exhibited by recent vision-language pre-trained models, this subsection focuses on three aspects: coarse-grained global matching, fine-grained local matching, and vision-language pre-trained models matching. Finally, it compares studies similar to our work.

Coarse-grained global matching

The process involves independently encoding images and texts into a unified embedding space using two separate branches, obtaining coarse-grained global representations through specific aggregators, and performing similarity-matching calculations. Existing work typically utilizes GCN (Li et al., 2019, 2024b; Wang et al., 2020b) or transformer (Wang et al., 2020a; Wen, Gu & Cheng, 2020; Wu et al., 2019) to enhance the contextual semantics of local features. For instance, Li et al. (2019) introduced a simple semantic reasoning network (VSRN) to learn local image features with key scene concepts. Wang et al. (2020a) introduced a consensus-aware module (CVSE) that incorporates common sense knowledge into local features of two patterns. An important research direction in coarse-grained global matching focuses on the work of clustering pooling strategies (Chen et al., 2021; Li et al., 2019; Zeng et al., 2021; Zhu et al., 2023). For instance, Li et al. (2019) leveraged gates and memory mechanisms to conduct global semantic inference on features, selectively identify discriminative information, and derive representations of the entire scene. Chen et al. (2021) introduced the optimal pooling method (GPO) that automatically adapts to different characteristics without manual tuning, down sampling intra-modal features into global features. Zhu et al. (2023) introduced a flexible ESA strategy, facilitating element-level fusion of image and text features in specific spatial dimensions. Kim, Kim & Kwak (2023) proposed the idea of utilizing a set that transforms an image or text sample into a collection of unique embedding vectors, designed to encompass diverse semantic elements of the samples. However, previous works still follow the existing paradigm, this work considers that the same feature may mean different things in different dimensions and have different importance in different dimensions. Aggregating vector weights in both different and same dimensions can maximize the acquisition of potentially rich dimensional semantics.

Fine-grained local matching

The goal is to identify all potential alignments between textual words and image regions, thereby inferring the comprehensive correlation between the image and text, better performance can usually be achieved (Chen et al., 2020; Diao et al., 2021; Fu et al., 2023; Lee et al., 2018; Liu et al., 2020; Wang et al., 2023b; Zhang et al., 2022a, 2022b, 2024b). Most work in this field is based on attention mechanisms. Lee et al. (2018) introduced the SCAN technique as a prominent approach for dynamically targeting the common semantics across different modalities in response to each query content. On this basis, many variants were proposed. The focus of a sequence of works is to devise precise and complex matches (Chen et al., 2020; Diao et al., 2021; Lee et al., 2018; Liu et al., 2020; Zhang et al., 2022b). For instance, Chen et al. (2020) introduced Recurrent Cross Attention, which iteratively extracts and enhances shared semantics across multiple levels. Zhang et al. (2022b) proposed measuring the degree of similarity and dissimilarity through two matches (NAAF). Pan, Wu & Zhang (2023) proposed a new encoding strategy (CHAN) to find the most relevant region-word pairs to address redundant alignment and cross-attention mechanisms. Diao et al. (2021) first used the vectorised representation method (SGRAF), which is able to calculate the similarity and inference between two modal at a finer granularity. Simultaneously, various techniques concentrates on leveraging external and other information to construct relationships within the same modal and between different modal of sentences and images (Fu et al., 2023; Zhang et al., 2022a). For example, Zhang et al. (2022a) proposed the region position (CMCAN) to reflect the surrounding visual scene, while Fu et al. (2023) emphasized and enriched the interaction dynamics between visual and texts through semantic relationships between instances. Nevertheless, the emphasis of these methods remains on purely calculating the localized similarity between image segments and textual terms to gather the overall image-text correlation, which can easily lead to excessive attention to the the connections between image areas and textual terms, thereby causing a discrepancy between the global and local information. our proposed method focuses on using coarse-grained global matching as a complement to fine-grained image segments and textual terms matching to solve the problem of semantic inconsistency between global and local information, and adopting cross-dimensional semantic dependency in the process of fine-grained image segments and textual terms matching to mine richer image and text features.

Vision-language pre-trained model matching

Early Vision-Language Pre-trained (VLP) models followed a detector-based roadmap (Chen et al., 2019). The CLIP model (Radford et al., 2021), ALIGN (Jia et al., 2021), and unicoder-VL (Li et al., 2020) respectively utilized 400 million image-text pairs, 1.8 billion noisy image-text pairs, and 3.8 million web image-text data for contrastive training. These models achieved efficient cross-modal retrieval and cross-modal understanding and generation tasks by integrating multimodal information. Although this approach demonstrated strong performance, it also posed challenges of expensive computation and unstable detector training. Subsequently, methods based on Vision Transformers (ViTs) adopted a pure Transformer architecture for image encoding, eliminating the need for object detectors and enabling an end-to-end VLP framework. For instance, Huang et al. (2021) proposed an “out-of-the-box” solution that does not rely on complex preprocessing steps; Kim, Son & Kim (2021) utilized the Transformer architecture to directly take unprocessed image segments and textual word embeddings as inputs, avoiding complex visual feature extraction processes. Li et al. (2021) introduced momentum distillation techniques, using a momentum model to enhance visual and language feature alignment, resulting in significant performance improvements across multiple vision-language tasks. However, these methods had limitations in fine-grained cross-modal alignment, increased computational costs, and introduced noisy visual information for cross-modal fusion. More recent works have made further advancements in enhancing the effectiveness of the pre-trained process for vision-language models. For instance, Fu et al. (2024) introduced a patch fusion method designed to obtain a compact representation of visual token sequences and strengthen the integration across different modes. Additionally, the BLIP-2 model series (Li et al., 2023) adopted a guided language-image pre-trained method utilizing pre-fixed image encoders and substantial language models, achieving effective fusion and precise matching of visual and language information.

Recent studies have produced works similar to this research. For instance, Xu et al. (2020) introduced the CASC joint framework that performs general local alignment and global semantic consistency multi-label prediction. Similarly, Li et al. (2024b) proposed the memory networks to achieve complementarity between independent and interactive embedding approaches, thereby improving retrieval efficiency. However, these methods do not consider address fine-grained cross-dimensional semantic matching, nor did it consider the relationship between different dimensions of the same feature and different features of the same dimension in coarse-grained matching. Zhang et al. (2023) conducted precise mining and modeling of dimensional semantic dependencies (x-dim), but still only performed fine-grained cross-dimensional semantic matching. In contrast, the approach presented in this work explicitly constructs a coarse-grained complementary cross-dimensional semantic aggregation network, performs fine-grained cross-dimensional semantic dependency matching, and further aggregates cross-dimensional image and text semantic to ensure consistency between global and local information.

Methodology

This work offers an in-depth elucidation of the proposed CDGCN, as depicted in Fig. 3. This work comprises four distinct components: (1) Feature extraction. (2) Fine-grained cross-dimensional semantic dependency branch learns interdependencies between different dimensions. (3) Coarse-grained cross-dimensional semantic aggregation module aggregates vector weights in different and same dimensions. (4) Global semantic matching is used as a supplement to local semantic alignment for coarse-fine-grained image-text matching.

Figure 3 Overview of the proposed CDGCN framework.

The proposed CDGCN framework combines coarse-grained cross-dimensional global semantic consistency with fine-grained local region word cross dimensional semantic alignment to quantify the correlation between sentences and images. CDGCN framework consists of two parts: (A) fine-grained cross-dimensional semantics dependency branch: Searching for semantic dependencies between different dimensions of vectors in a similarity shared space for fine-grained word and image region matching. (B) Coarse-grained cross-dimensional semantic aggregation (CGDSA) module: consider the varying significance of different dimensions within the same feature, based on the interaction between different vectors in the same dimension, cross-dimensional aggregation weights are used to aggregate local fine-grained features into global features.

Feature extraction

Image representation. For an image v, this work uses a bottom-up attention network (Anderson et al., 2018) to extract salient regions, which employs ResNet-101 (He et al., 2016) as the backbone and is implemented by Faster-RCNN (Ren et al., 2015). Then, this work adds a fully connected (FC) layer to map each region feature to a d-dimensional local feature, represented as V={v1,...vm}∈Rm×d, which is the visual fragment and local feature of image v, and the m is the number of image region features.

Text representation. For a sentence u, this work uses a sequence model, bidirectional gated recursive unit (BiGRU) (Schuster & Paliwal, 1997), or pre-trained BERT model (Devlin et al., 2018) to extract the word feature set. This work also adds a fully connected (FC) layer to maintain the same dimension as the image, represented as U={u1,...un}∈Rn×d, which is the text fragment and local feature of the text u, n is the number of word features.

Fine-grained cross-dimensional semantic dependency branch

For the semantic similarity between any word feature u={ui}i=1d and region feature v={vi}i=1d, according to Fig. 1B, typical approaches often employ implicit independent aggregation to reflect it for all dimension correspondences, i.e., ∑i=1d⁡si, where si can be determined by the product of scalars ui and vi in internal product operations. That is to say, The aggregated process of cross-modal correspondence vectors si={si}i=1d in the similarity sharing-space is based on the unconscious assumption that each dimension is perceived as a distinct isolated component.

Inspired by Zhang et al. (2023), not all dimensions in a similarity shared-space are independent, and some local dimensions have potential relationships between them, defined as cross-dimensional semantic dependencies. This work introduces the conditional probability of each dimension to represent joint semantics. It selects dimensions with significant correlation for sparse correlation probability learning based on adaptive adjustment of learned weights to sparsity. Firstly, as shown in the dimension-dependent perception aggregation in Eq. (1):

(1) Aggw(s)=∑i=1d⁡||[w11⋯w1d⋮⋱⋮wd1⋯wdd]W×[s1,...,si,....]T||l2

where W∈Rd×drepresents a learning weight matrix and each row of which represents a semantic dependency between various dimensions. Specifically, for row i of the weight matrix, The formula for si^ is as follows:

(2) s^i=wi1.s1+...+wij.sj+...+wid.sd,j∈[1,d].

It models the dependency relationships among all dimensions within the shared representation space of the i-th joint representation, collectively representing latent semantics. To explicitly quantify, the conditional probabilities for each dimension are introduced as follows:

(3) p(s^i|sj)=Sigmoid(wij),j∈[1,d],

where p(si^|sj)∈[0,1] reflects the extent of dependency of the j-th dimension for the jointly represented s^i. The likelihood that this dimension in the shared space participates in the joint representation is positively correlated with the value p(s^i|sj). The learned conditional probability {p(s^i|sj)}j=1d represents the dependence of all dimensions on latent semantics s^i, Zhang et al. (2023) proved in the article that the probability density is approximately a normal distribution. Drawing from the statistical characteristics of conditional probability, this work initially allows the model to independently acquire a soft threshold for distinguishing if the dimensions exhibit joint dependencies. The significance is as follows:

(4) ti=ui+αi⋅σi

where ui and σi denote the mean and standard deviation of the sampling probability values {p(s^i|sj)}j=1d. αi is a learnable parameter for adaptively adjusting the soft threshold to regulate the ratio of selected dimensions. Further scaling operations obtain the modified sparse probability as:

(5) p^(s^i|sj)=δ(lγ(p(si^|sj)−ti))

where γ represents a trainable scaling parameter, δ(⋅) is the activation function.

In other words, based on the learning of conditional probability p(s^i|sj), ti as a critical point, dimensions with dependency less than the threshold ti will be discarded, while dimensions with dependency greater than the threshold ti will continue to be used. With the operations mentioned above, Eq. (1) will be modified to Eq. (6):

(6) si^=∑i=1d⁡p^(si^|sj)⋅wij⋅sj,j∈[1,d].

Finally, adaptive dimensional semantic dependency perception can be formulated as:

(7) Aggw(s)=∑i=1d⁡||(P^⊙W)×sT||l2

where P^=p^(si|s^j), i,j∈[1,d] can be regarded as an adaptive regularizer to achieve fine-grained dimensional dependency modeling.

Coarse-grained cross-dimensional semantic aggregation module

The fine-grained cross-dimensional semantic dependency branch has completed the fine-grained cross-dimensional region-word similarity computation, and in order to avoid excessive focus on the local image region and text word information to maintain the consistency of global and local semantic information, coarse-grained global image text information needs to be introduced as an auxiliary balance. However, current global features generated by aggregation of commonly used pooling operations, such as average pooling and maximum pooling, for example, contain a lack of semantic features. Inspired by Wu et al. (2023), his work found that different dimensions may have different meanings for the same feature. Taking the term “match” as an illustration, certain dimensions may connote “light” while others represent “competition”. Similarly, some dimensions of “doctor” may represent “physician” while others represent “doctor of philosophy”. It means that when consolidating global features, the significance of various dimensions within the same feature vector should vary.

Based on previous research, this work introduces a novel aggregation module called CGDSA. Unlike common pooling operations, as shown in Fig. 4, CGDSA module considers the relationships between different vectors on the same dimension and the differences of the same feature between different dimensions. Specifically, taking an image as an example, this work first enhances the obtained features:

(8) F=W2(ReLu(W1U+b1))+b2

where W1, b1 and W2, b2 are the parameters of the two fully connected layers, the fully connected layer transforms u into the d-dimensional common space to standardize the representation dimension, ReLu activation function for nonlinear mapping to obtain higher semantic information.

Figure 4 The structural partitioning of coarse-grained cross-dimensional semantic aggregation (CGDSA) module.

(1) Same-Dimension Attention Aggregation (SDAA). (2) Different-Dimension Aggregation (DDA).

Then, this work draws on an additional external d * d space (denoted by M) to keep the image and text mapped within a fixed dimensional attention space, which allows for relationships between different vectors in the same dimension. The expression formula for this matrix is as follows:

(9) F1=softmax(F∗M).

Meanwhile, under the influence of the combination of vertical vectors, discriminative information in the horizontal dimension cannot be flexibly integrated in a fine-grained manner. Therefore, this work aggregates features from the direction of column elements to obtain aggregation weights to obtain richer features, and the specific formula is as follows:

(10) F2=softmax(F).

Note that the softmax function is executed separately in the region dimension column direction. Column-wise softmax involves applying the softmax function to each column individually, ensuring that the sum of N elements in each column equals 1.

Additionally, this work introduces vectors as aggregation weights, which can achieve finer granularity aggregation compared to scalars as aggregation weights. Lastly, this work amalgamates features of the identical dimension mapped to the external space and features of diverse dimensions to derive the ultimate aggregated feature. The computation formula is shown as Eq. (11):

(11) u^=∑i=1N⁡F1⊙(ui)+∑i=1N⁡F2⊙(ui).

Note that F1, F2 are the vectors, where each element stands for the combined weight of each dimension of ui, of course, the aggregation of textual features can also be accomplished by the operation of this module, notated as v^.

Coarse-fine-grained semantic matching

The goal of the CDGCN method is the combination of fine-grained cross-dimensional semantic dependency match branch and coarse-grained cross-dimensional semantic aggregation match branch. The CDGCN is included into the cross attention inference to establish cross-modal fine-grained matching. Taken out an image-text pair ( U,V) from the dataset which comprises n text-words {uq}q=1n and m salient image-regions {vp}p=1m, the semantic similarity of all text-word image-region pairs is obtained as follows (Lee et al., 2018):

(12) rqp=Agg(sqp),s.t.sqp=uq⊙vp.

For each text word query uq, its relevant area can be represented as:

(13) v^q=∑p=1m⁡βqpvp,s.t.βqp=exp(λ⊙σ(rqp))∑p=1m⁡exp(λ⊙σ(rqp))

where βqp is weight, λ denotes a trainable scaling parameter. Following the same logic, this work can obtain the correlation score between the q-th text-word and the image as: r^q=Agg(sq), s^q=uq⊙v^q. Finally, the overall correlation between image and text is inferred as:

(14) S(U,V)fine=1n∑q=1n⁡r^q.

Similarly, this work maps the aggregated image features v^ and text features u^ into a shared space to compute cosine similarity for coarse-grained global matching: s(v^,u^)coarse=v^⊤⋅u^||v^||⋅||u^||.

This work trains and computes bi-directional ranking losses (Faghri et al., 2017) for matching image-text pairs:

(15) L(U,V)=[γ−S(U,V)+S(U,V−)]++[γ−S(U,V)+S(U−,V)]+

The final objective function of the CDGCN method derived is:

(16) L=L(U,V)fine+αL(U,V)coarse

where V−denotes hardest negative image and U−denotes hardest negative text, α is a hyper-parameter that balances the loss functions between fine-grained cross-dimensional semantic matching and coarse-grained cross-dimensional semantic aggregation matching.

Results and analysis

Datasets and evaluation metrics

Datasets: This work uses Flickr30K dataset (Young et al., 2014) and the MS-COCO dataset (Lin et al., 2014), where each image is paired associated with five texts descriptions. The Flickr30k dataset comprises 29,000 training images, 1,000 testing images and 1,014 validation images. The MS-COCO contains 82,738 training images, 5,000 testing images, and 5,000 validation images. Results for MS-COCO were evaluated by averaging outcomes across five runs for the 1 K test images and for the entire set of 5K test images. The Flickr30K dataset can be accessed at: https://shannon.cs.illinois.edu/DenotationGraph/; and the MSCOCO dataset is accessed through URL: https://cocodataset.org/#download. Codes are accessible at https://github.com/meizhenliu/CDGCN-code.

Evaluation metrics: The proposed method’s effectiveness is assessed using metrics R @ K (where K = 1, 5, 10) and rSum (Chen et al., 2021). Specifically, R@K quantifies the proportion of accurately retrieved true results among the top K items returned, whereas rSum represents the sum of all R@K scores. Particularly, R@K indicates the percentage of correct ground truth results retrieved within the first K search items, and rSum signifies the total of all R@K values.

Implementation details

All experiments have been conducted using an NVIDIA GeForce RTX 3090 GPU and the proposed CDGCN is implemented using PyTorch. In the visual branch, this work uses the Bottom-Up and Top-Down Attention (BUTD) network (Anderson et al., 2018; Li et al., 2019) pre-trained on the Visual Genome (VG) dataset to extract N = 36 region features from images, each with 2,048 dimensions. These features are then sent to a linear fully connected layer to map them into a shared space of 1,024 dimensions. For text representation, this work employs BiGRU model with a word embedding size of 300, along with the default pre-trained BERT (Devlin et al., 2018) with 768 dimensions. When using the BiGRU text encoder, this work initialize the word vectors with 300-dimensional global vectors (GLoVe) embeddings (Pennington, Socher & Manning, 2014), applying a dropout rate of 0.1 during training to enhance the generalization of text representations. In the case of using the BERT text encoder, our work makes use of the standard pre-trained BERT model, consisting of 12 transformer layers, each equipped with 12 attention heads and a hidden layer dimension of 768 units. This work also uniformly applys text augmentation strategies to train BERT. Additionally, after encoding both the images and sentences, this work incorporates L2 normalization to process the features, ensuring the stability and consistency of the representations.

During the training process, the visual encoder’s parameters remain unchanged, whereas the parameters of BERT undergo fine-tuning. This work uses the Adam optimizer, with an initial learning rate of 0.0002 for the Flickr30k dataset and 0.0005 for the MSCOCO dataset. The training process spans 30 epochs across both datasets, with a batch size of 128 for each dataset. The learning rate is reduced by 10% every 15 epochs. The margin parameter λ is set to 0.2. The hyperparameter α, which balances the coarse-grained cross-dimensional semantic aggregation matching branch and the fine-grained cross-dimensional semantic dependency matching branch in the loss function, is set to 0.06 for the Flickr30k dataset and 0.1 for the MSCOCO dataset.

During testing, this work evaluates the single model obtained from training, denoted as-single. For model ensembling, this work averages the similarity matrices of any two models obtained from multiple training runs, the ensembled results are denoted as-ensembled*.

Benchmark with state-of-the-art methods

The performance of the CDGCN is compared with state-of-the-art models using individual and integrated models. In this section, the CDGCN which is a relatively low-cost small model is compared with previous low-cost small models. In addition, this work also presents the retrieval results of the state-of-the-art vision-language pre-training large models (Li et al. 2023, 2021) in Table 1. It can be seen that the vision-language pre-training large models (such as BLIP-2) demonstrate superior ability with significant improvement over existing methods. To ensure an unbiased comparison of experimental results, this work categorizes existing methods (focusing only for the low-cost small models) into two groups based on their distinct approaches to extracting visual and textual backbone features. Tables 1 and 2 respectively showcase the quantitative findings of the CDGCN and existing methods on the Flickr30K and MSCOCO 1 K datasets. Even when employing a standalone model rather than an integrated one, the proposed CGDSN has shown significant improvement compared to the state-of-the-art methods (Fu et al., 2023; Pan, Wu & Zhang, 2023; Yao et al., 2024; Zhang et al., 2023; Zhu et al., 2023). The proposed CDGCN has impressive advantages in R @ 1,5,10 and rSum. Notably, compared to the existing SOTA X-dim method (Zhang et al., 2023), the proposed CDGCN achieved a rSum improvement rate of 4.6–7.7% relative to the two datasets. For larger MSCOCO 5K test sets, the proposed method surpasses the SOTA model in all evaluation metrics and has a significant performance gap in Table 3. The proposed CDGCN achieved a rSum improvement rate of 5.0–8.8% for the two datasets, which more convincingly demonstrates the superiority of the proposed CDGCN.

Table 1 The comparisons with state-of-the-art methods on Flickr30K test set, utilizing comprehensive assessment indicators such as recall at K (R@K, where K is 1, 5, 10, representing the recall rate of the top K results relative to the query) and rSum, which is the sum of recall-K values at K = 1, 5, 10 for both image retrieval based on text and text retrieval based on image tasks.

The term “Region” indicates the use of Faster-RCNN to extract features from image regions. “BiGRU” and “BERT” signify their use in extracting word features from texts. “Single” refers to the result obtained from a single model, while “ensemble*” indicates the combined results of two models.

Method	IMG → TXT	TXT → IMG		
R@1R@5R@10 (%)	R@1R@5R@10 (%)	rSum (%)	
Vision-language Pre-trained (VLP) models	
CLIP (Radford et al., 2021)	88.0	98.7	99.4	68.7	90.6	95.2	540.6	
UNITER (Chen et al., 2019)	83.6	95.7	97.7	68.7	89.2	93.9	528.8	
ALBEF (Li et al., 2021)	94.1	99.5	99.7	82.8	96.3	98.1	570.5	
BLIP-2 ViT-g (Li et al., 2023)	97.6	100.0	100.0	89.7	98.1	98.9	584.3	
LAPS (Fu et al., 2024)	85.1	97.7	99.2	74.0	93.0	96.3	545.3	
Region + BiGRU	
SCAN* (Lee et al., 2018)	67.4	90.3	95.8	48.6	77.7	85.2	465.0	
ADAPT* (Wehrmann, Kolling & Barros, 2020)	76.6	95.4	97.6	60.7	86.6	92.0	508.9	
SHAN* (Ji, Chen & Wang, 2021)	74.6	93.5	96.9	55.3	81.3	88.4	490.0	
AME (Li, Niu & Zhang, 2022)	74.9	93.5	97.0	58.9	84.7	90.2	499.2	
BiKA* (Zhu et al., 2022)	75.2	91.6	97.4	54.8	82.5	88.6	490.1	
CMCAN (Zhang et al., 2022a)	77.5	94.3	96.9	58.8	82.9	88.9	499.3	
NAAF (Zhang et al., 2022b)	79.6	96.3	98.3	59.3	83.9	90.2	507.6	
ESA (Zhu et al., 2023)	82.6	95.9	98.1	61.1	85.9	91.1	514.7	
CHAN (Pan, Wu & Zhang, 2023)	79.7	94.5	97.3	60.2	85.3	90.7	507.8	
HREM (Fu et al., 2023)	79.5	94.3	97.4	59.3	85.1	91.2	506.8	
The proposed CDGCN-single	79.3	95.6	97.6	61.3	85.6	91.2	510.6	
The proposed CDGCN- ensemble*	81.4	95.9	97.9	63.2	86.9	92.2	517.5	
Region + Bert	
CAMERA* (Qu et al., 2020)	78.0	95.1	97.9	60.3	85.9	91.7	508.9	
DSRAN* (Wen, Gu & Cheng, 2020)	77.8	95.1	97.6	59.2	86.0	91.9	507.6	
GPO (Chen et al., 2021)	81.7	95.4	97.6	61.4	85.9	91.5	513.5	
AME (Li, Niu & Zhang, 2022)	77.1	95.1	97.3	61.2	86.1	91.4	508.3	
VSRN++ (Li et al., 2022a)	79.2	94.6	97.5	60.6	85.6	92.3	508.9	
MV-VSE (Li et al., 2022c)	82.1	95.8	97.9	63.1	86.7	93.1	517.5	
CHAN (Pan, Wu & Zhang, 2023)	80.6	96.1	97.8	63.9	87.5	92.6	518.5	
X_DIM (Zhang et al., 2023)	83.6	96.0	98.4	65.0	88.7	93.1	524.8	
RAAN* (Yao et al., 2024)	77.1	93.6	97.3	56.0	82.4	89.1	495.5	
The proposed CDGCN-single	83.7	96.6	99	65.7	88.4	93.2	526.7	
The proposed CDGCN- ensemble*	84.3	96.6	99.0	68.5	90.0	94.1	532.5	

Table 2 Comparisons with state-of-the-art methods on MSCOCO 1K test set, utilizing comprehensive assessment indicators such as recall at K (R@K, where K is 1, 5, 10, representing the recall rate of the top K results relative to the query) and rSum, which is the sum of recall-K values at K = 1, 5, 10 for both image retrieval based on text and text retrieval based on image tasks.

The term “Region” indicates the use of Faster-RCNN to extract features from image regions. “BiGRU” and “BERT” signify their use in extracting word features from texts. “single” refers to the result obtained from a single model, while “ensemble*” indicates the combined results of two models.

Method	IMG → TXT
R@1R@5R@10(%)	TXT → IMG
R@1R@5R@10(%)	rSum(%)	
Region + BiGRU	
SCAN* (Lee et al., 2018)	73.4	93.8	97.8	57.5	87.3	94.3	504.1	
ADAPT* (Wehrmann, Kolling & Barros, 2020)	76.5	95.6	98.9	62.2	90.5	96.0	519.8	
SHAN* (Ji, Chen & Wang, 2021)	76.8	96.0	98.7	62.6	89.6	95.8	519.8	
AME (Li, Niu & Zhang, 2022)	77.1	95.4	98.3	62.8	89.6	95.8	519.8	
BiKA* (Zhu et al., 2022)	77.6	96.5	98.6	62.8	89.4	95.1	518.1	
CMCAN (Zhang et al., 2022a)	79.7	96.6	98.8	63.3	90.4	96.2	525.2	
NAAF (Zhang et al., 2022b)	78.1	96.1	98.6	63.5	89.6	95.3	521.2	
ESA (Zhu et al., 2023)	79.6	96.5	98.7	63.5	90.9	96.1	525.3	
CHAN (Pan, Wu & Zhang, 2023)	79.7	96.7	98.7	63.8	90.4	95.8	525.0	
HREM (Fu et al., 2023)	80.0	96.0	98.7	62.7	90.1	95.4	522.8	
The proposed CDGCN-single	79.6	96.5	98.6	63.9	90.7	96.0	525.3	
The proposed CDGCN-ensemble*	81.2	96.9	98.7	65.3	91.2	96.3	529.6	
Region+BERT	
CAMERA* (Qu et al., 2020)	77.5	96.3	98.8	63.4	90.9	95.8	522.7	
DSRAN* (Wen, Gu & Cheng, 2020)	78.3	95.7	98.4	64.5	90.8	95.8	523.5	
GPO (Chen et al., 2021)	79.7	96.4	98.9	64.8	91.4	96.3	527.5	
AME (Li, Niu & Zhang, 2022)	78.5	96.1	98.7	63.7	90.1	95.6	522.7	
VSRN++ (Li et al., 2022a)	77.9	96.0	98.5	64.1	91.0	96.1	523.6	
MV-VSE (Li et al., 2022c)	80.4	96.6	99.0	64.9	91.0	96.0	528.1	
CHAN (Pan, Wu & Zhang, 2023)	81.4	96.9	98.9	66.5	92.1	96.7	532.6	
X_DIM (Zhang et al., 2023)	82.2	97.2	99.1	66.9	92.0	96.6	534.0	
RAAN* (Yao et al., 2024)	76.8	96.4	98.3	61.8	89.5	95.8	518.6	
The proposed CDGCN-single	82.9	97.2	98.9	67.2	92.3	96.8	535.3	
The proposed CDGCN-ensemble*	84.2	97.4	99.1	69.0	93.1	97.2	539.9	

Table 3 Comparisons with state-of-the-art methods on MSCOCO 5K test set, utilizing comprehensive assessment indicators such as recall at K (R@K, where K is 1, 5, 10, representing the recall rate of the top K results relative to the query) and rSum, which is the sum of recall-K values at K = 1, 5, 10 for both image retrieval based on text and text retrieval based on image tasks.

The term “Region” indicates the use of Faster-RCNN to extract features from image regions. “BiGRU” and “BERT” signify their use in extracting word features from texts. “Single” refers to the result obtained from a single model, while “ensemble*” indicates the combined results of two models.

Method	IMG → TXT
R@1R@5R@10(%)	TXT → IMG
R@1R@5R@10(%)	rSum(%)	
Region+BiGRU	
IMRAM* (Chen et al., 2020)	53.7	83.2	91.0	39.7	69.1	79.8	415.5	
MV-VSE (Li et al., 2022c)	56.7	84.1	91.4	40.3	70.6	81.6	424.6	
AME (Li, Niu & Zhang, 2022)	54.0	82.1	90.7	40.1	70.2	80.5	417.6	
ESA (Zhu et al., 2023)	58.2	84.8	91.8	41.2	71.4	82.2	429.6	
NAAF* (Zhang et al., 2022b)	58.9	85.2	92.0	42.5	70.9	81.4	430.9	
X-DIM (Zhang et al., 2023)	59.3	86.0	92.5	43.2	71.9	82.4	435.3	
CHAN (Pan, Wu & Zhang, 2023)	60.2	85.9	92.4	41.7	71.5	81.7	433.4	
The proposed CDGCN-single	58.1	85.5	92.0	42.3	71.7	81.9	431.4	
The proposed CDGCN-ensemble*	60.5	86.7	92.8	43.8	72.8	82.7	439.4	
Region+BERT	
CAMERA* (Qu et al., 2020)	55.1	82.9	91.2	40.5	71.7	82.5	423.9	
DIME (Qu et al., 2021)	59.3	85.4	91.9	43.1	73.0	83.1	435.8	
AME (Li, Niu & Zhang, 2022)	57.1	83.5	91.6	42.2	71.7	82.0	428.1	
VSRN++* (Li et al., 2022a)	54.7	82.9	90.9	42.0	72.2	82.7	425.4	
MV-VSE (Li et al., 2022c)	59.1	86.3	92.5	42.5	72.8	83.1	436.3	
X-DIM (Zhang et al., 2023)	62.7	87.0	93.3	45.1	73.6	84.4	445.1	
CHAN (Pan, Wu & Zhang, 2023)	59.8	87.2	93.3	44.9	74.5	84.2	443.9	
The proposed CDGCN-single	65.1	87.9	93.8	45.9	74.9	84.6	452.3	
The proposed CDGCN-ensemble*	67.5	89.2	94.5	47.7	76.6	85.8	461.1	

This research achievement substantially advances the acquisition and understanding of image and text features, enhancing the robustness and the accuracy of image-text matching. It is well suited for application across various multimodal domains, including e-commerce, intelligent assistants, medical image analysis, and social media. This technology enables more accurate image retrieval and recommendation systems, improves user interaction experiences, optimizes content generation and editing tools, and supports intelligent diagnosis and public opinion analysis systems. Hence, this work improves the operational efficiency and user satisfaction of various industries and promotes innovation and development of multimodal data processing technology.

Ablation studies

As shown in Tables 4 and 5, this work conducts comprehensive ablation studies on the Flick30k dataset to validate the efficacy of the proposed CDGCN components. By default, experiments are performed on the fusion mechanism framework using Region+BERT.

Table 4 The ablation study of coarse-grained cross-dimension aggregation branches and fine-grained cross-dimension dependency branches on Flickr30K, using comprehensive evaluation metrics including recall at K (R@K, K = 1, 5, 10, defined as the recall rate at the top K results to the query) and rSum, which sums the recall-K values at K = 1, 5, 10 for both image-to-text and text-to-image retrieval tasks.

Using Faster-RCNN to extract region features for images and use BERT to extract word features for texts. “Single” denotes the result of one model,“ensemble*” denotes the ensemble results of two models.

Method	IMG → TXT
R@1R@5R@10(%)	TXT → IMG
R@1R@5R@10(%)	rSum(%)	
W/O fine-grained branch	82.1	95.6	97.6	63.5	87.7	92.6	519.1	
W/O fine-grained cross-dimension	65.3	89.5	94.5	45.5	77.1	85.7	457.6	
W/O coarse-grained branch	83.6	96.0	98.4	65.0	88.7	93.1	524.8	
The proposed CDGCN-single	83.7	96.6	99	65.7	88.4	93.2	526.7	
The proposed CDGCN-ensemble*	84.3	96.6	99.0	68.5	90.0	94.1	532.5	

Table 5 The ablation study of Flickr30K coarse-grained cross-dimensional aggregation branch, using comprehensive evaluation metrics including recall at K (R@K, K = 1, 5, 10, defined as the recall rate at the top K results to the query) and rSum, which sums the recall-K values at K = 1, 5, 10 for both image-to-text and text-to-image retrieval tasks.

Using Faster-RCNN to extract region features for images and use BERT to extract word features for texts. “Single” denotes the result of one model, “ensemble*” denotes the ensemble results of two models.

Method	IMG → TXT
R@1R@5R@10(%)	TXT → IMG
R@1R@5R@10(%)	rSum(%)	
Only same-dimension attention aggregation (SDAA)	82.3	96.0	98.4	64.4	87.7	93.0	521.9	
Only different-dimension aggregation (DDA)	82.1	96.3	98.1	65.5	88.9	93.4	524.3	
With scalar	55.7	83.1	90.6	41.0	71.6	80.8	422.9	
GPO	83.6	95.7	98.0	64.3	87.8	92.6	522.0	
Average pool	67.0	89.9	94.7	53.3	81.7	88.7	475.3	
Max pool	67.5	90.0	94.6	47.8	78.5	86.6	465.0	
The proposed CDGCN-single	83.7	96.6	99	65.7	88.4	93.2	526.7	
The proposed CDGCN-ensemble*	84.3	96.6	99.0	68.5	90.0	94.1	532.5	

Firstly, this work conducts ablation studies to explore the efficacy of two branches of the CDGCN. The without (W/O) fine-grained branch represents the removal of fine-grained cross-dimensional semantic dependency matching branches, the W/O fine-grained cross-dimension branch represents the retention of fine-grained semantic matching branches without performing similarity cross-dimensional dependency calculations, and the W/O coarse-grained cross-dimensional semantic matching branch represents the removal of coarse-grained cross-dimensional semantic matching branches. The comparative outcomes are depicted in Table 4. The performance of rSum without fine-grained cross-dimension module decreased by 4.8%, indicating the importance of using fine-grained cross-dimension semantic matching branches. In addition, the performance of rSum W/O fine-grained cross-dimension shows a significant decrease, which also indirectly verifies the powerful effect of similarity cross-dimension dependency. The performance of the rSum W/O coarse-grained branch module decreased by 1.9%, indicating that using coarse-grained cross-dimensional semantic matching branches can effectively obtain rich image text features while maintaining consistency between overall and local features, thereby enhancing the performance of image-text retrieval.

Secondly, to verify the effectiveness of the CGDSA module, this work conducts ablation experiments on each component in the image-text retrieval task. It uses three standard pooling methods to aggregate local features and Table 5 presents the comparative outcomes. This work shows that only same dimension attention aggregation (OSDAA) represents only external spatial attention aggregation in the same dimension, meanwhile, only different dimension attention aggregation (ODDA) represents only cross-dimensional aggregation in different dimensions with scalar representing scalar aggregation weight, and row-wise softmax representing row element direction aggregation feature. As the data displayed in Table 5, it is clear that the importance of each component in CGDSA module is significant. If only the DDA module is used, without the SDAA module and cross-dimensional aggregation features of different dimensions, the performance of rSum decreases by 4.8%. If only the SDAA module is used, without the DDA module, and without features that interact with different vectors of the same dimension, the performance of rSum decreases by 2.5%. Using scalar aggregation weights instead of vectors cannot obtain richer element-level aggregation, significantly reducing rSum’s performance. Furthermore, the CGDSA method performs better than other methods. For example, using mean pooling resulted in a 12.5% and 11.4% decrease in R @ 1 for text retrieval and image retrieval, individually. GPO resulted in a 2.1% and 1.8% decrease in R @ 1 for text retrieval and image retrieval, particularly, resulting in a 6.4% decrease in rSum performance. Different from simple pooling methods, this work emphasizes the differences in feature dimensions and the mapping of identical dimensions and vectorizes the aggregation weights, thereby achieving a finer-grained aggregation of features across identical and distinct dimensions.

In addition, as a plug-and-play CGDSA module, the CGDSA method can also be directly used in visual semantic embedding (VSE) tasks. Compared with other pooling strategies, all indicators of the proposed CGDSA method have been significantly improved. Table 6 illustrates that the CGDSA module enhances the effectiveness of visual semantic embedding. The CGDSA method improves R @ 1 by 12.5% and 11.4% respectively when compared to average pooling text retrieval and image retrieval. In addition, the improvement of R@1 using the CGDSA method has improved by 2.1% and 1.8%, particularly, resulting in a 6.4% performance improvement when compared to the GPO method. In addition, this work also compared the fusion methods of only SDAA and only DDA, and the results showed that the sum of the elements in the SDAA and DDA had the best effect. The CGDSA (concatenation) indicates that the work performed subsequent operations on elements; meanwhile, CGDSA (element added) suggests that the work directly added elements to operate. The CGDSA (element added) obtains the most encouraging results compared to other methods since the proposed method managed to extract richer information.

Table 6 Ablation study on visual semantic embedding using different pooling strategies, using comprehensive evaluation metrics including recall at K (R@K, K = 1, 5, 10, defined as the recall rate at the top K results to the query) and rSum, which sums the recall-K values at K = 1, 5, 10 for both image-to-text and text-to-image retrieval tasks.

Using Faster-RCNN to extract region features for images and use BERT to extract word features for texts.

Method	IMG → TXT
R@1R@5R@10(%)	TXT → IMG
R@1R@5R@10(%)	rSum(%)	
Only SDAA	82.1	94.9	97.5	56.2	84.4	90.6	505.7	
Only DDA	79.7	94.3	97.1	56.9	84.8	91.3	504.0	
GPO	81.2	95.7	97.9	62.3	86.5	92.3	515.9	
Average pool	70.8	90.3	96.3	52.7	81.2	88.5	478.9	
Max pool	81.5	95.3	97.7	61.4	85.9	91.8	513.6	
Fusion method	
The CGDSA (Concatenation)	80.7	94.7	96.5	56.5	83.9	90.4	502.7	
The proposed CGDSA (Element added)	83.3	95.8	97.9	64.1	88.3	92.8	522.3	

Sensitivity to parameters

Hyperparameter α: After computing the coarse-grained cross-dimensional semantic aggregation matching branch and the fine-grained cross-dimensional semantic dependency matching branch, it is imperative to combine the loss functions of these two branches to ensure the consistency of local and global semantic information. The hyperparameter α serves as a weighting factor to balance these loss functions. Specifically, α adjusts the relative contribution of the two branches to the overall loss, as represented in Eq. (16). Typically α is a scalar value, ranging between 0 and 1. Within the CDGCN framework, α plays a crucial role in balancing the loss of the coarse and fine-grained matching branches. By defining α and dynamically adjusting it during the training process, the optimal parameter value that maximizes the model’s performance can be discovered. Other parameters of the CDGCN remain unchanged during the adjustment of α.

The findings of the ablation experiments are presented in Tables 7, 8, and Fig. 5. When α is set to 0, it indicates that the coarse-grained cross-dimensional global matching branch is not engaged in the calculation, resulting in relatively low performance of CDGCN. Notably, setting α to 0.1 on Flick30k and 0.05 on MSCOCO yields the optimal performance of CDGCN. Conversely, as α rises, the retrieval performance of CDGCN progressively deteriorates. The performance degradation is attributed to an excessive emphasis on global semantic information from images and text, which undermines the effective matching of detailed features in images regions and text words. This imbalance affects the performance of the model and compromises the model’s efficacy. Hence, selecting an appropriate α is crucial for CDGCN. Ultimately, when all parameter values are varied within appropriate ranges, the performance remains relatively stable. This demonstrates the robustness of the CDGCN and its insensitivity to the selection of hyper-parameters.

Table 7 The value of hyper-parameter α on Flick30k, using comprehensive evaluation metrics including recall at K (R@K, K = 1, 5, 10) and rSum, which sums the recall-K values at K = 1, 5, 10 for both image-to-text and text-to-image retrieval tasks.

Using Faster-RCNN to extract region features for images and use BERT to extract word features for texts.

α	IMG → TEXT(%)
@1 @5 @10	TEXT → IMG(%)
@1 @5 @10	rSum(%)	
1	29.8	58.7	71.4	19.4	45.2	58.2	282.7	
0.5	55.7	82.1	89.7	41.6	69.8	79.3	418.2	
0.1	83.7	96.6	99.0	65.7	88.4	93.2	526.7	
0.06	84.8	96.6	98.5	64.8	87.7	93.2	525.6	
0.03	83.2	96.1	98.1	62.9	87.4	92.2	520.0	
0	83.0	96.0	97.2	65.0	88.7	93.1	523.0	

Table 8 The value of hyper-parameter α on MSCOCO, using comprehensive evaluation metrics including recall at K (R@K, K = 1, 5, 10) and rSum, which sums the recall-K values at K = 1, 5, 10 for both image-to-text and text-to-image retrieval tasks.

Using Faster-RCNN to extract region features for images and use BERT to extract word features for texts.

α	IMG → TEXT
@1 @5 @10(%)	TEXT → IMG
@1 @5 @10(%)	rSum(%)	
MSCOCO 1k	
1	34.2	67.3	81.6	23.7	58.7	76.1	341.6	
0.5	39.8	71.8	84.1	28.3	63.8	79.6	364.7	
0.1	68.7	91.6	96.6	49.9	82.3	91.3	480.4	
0.05	82.9	97.2	98.9	67.2	92.3	96.8	535.2	
0.03	82.0	97.2	99.0	67.0	92.4	97.0	534.7	
0	81.3	97.2	98.6	65.2	91.3	96.3	529.9	
MSCOCO 5k	
1	15.1	36.1	50.4	9.4	27.2	39.6	177.7	
0.5	18.5	43.1	56.6	11.8	32.0	45.4	207.4	
0.1	56.1	82.5	90.2	39.6	69.8	80.4	418.4	
0.05	65.1	87.9	93.8	45.9	74.9	84.6	452.3	
0.03	64.0	87.0	93.5	45.3	74.6	84.4	448.8	
0	62.7	87.0	93.3	45.1	73.6	84.4	445.1	

Figure 5 Impact of hyper-parameters. as α changes, the changes of metrics R @ K and rSum.

(A) on Flick30k, (B) on MSCOCO 1k, (C) on MSCOCO 5 k.

Generalization study

In cross-modal image-text matching tasks, evaluating a model’s generalization capabilities is critical for ensuring its practicality in real-world applications. The significance of cross-dataset generalization research lies in its potential to enhance the robustness of models across disparate datasets, thereby improving the efficiency of real-world deployments. Assessing model’s performance across multiple datasets offers significant insights into the model’s adaptability to varying environmental conditions–a key factor in scenarios where data diversity is unavoidable. Strengthening this robustness minimizes dependence on specific datasets and significantly enhances the model’s generalization ability, ensuring reliable performance on unseen data and increasing its practical applicability.

This work conducts cross-validation experiments to verify the proposed CDGCN’s generalization ability for overall coarse-fine-grained image-text matching. The experiment involves training baseline models and CDGCN on the MSCOCO dataset, followed by cross-validation on the Flickr30K dataset to test the CDGCN’s zero-shot ability. The detailed outcomes are depicted in Table 9, whether it’s the “Region+BiGRU” version or the “Region+Bert” version, the CDGCN model exhibits stronger generalization capabilities than the baseline model, demonstrating more robust performance. The findings validate the efficacy of the proposed CDGCN in cross-dimensional semantic aggregation and coarse-grained semantic consistency.

Table 9 Comparison of model generalization capabilities, using comprehensive evaluation metrics including recall at K(R@K, K = 1, 5, 10) and rSum, which sums the recall-K values at K = 1, 5, 10 for both image-to-text and text-to-image retrieval tasks, Region represents using Faster-RCNN to extract region features for images.

BiGRU and BERT represent using them to extract word features for texts. Baseline denotes the basic image-text matching model.

Method	IMG → TEXT(%)
@1 @5 @10	TEXT → IMG(%)
@1 @5 @10	rSum(%)	
Region+BiGRU	
Baseline	68.8	91.3	95.6	53.0	78.6	85.6	472.9	
CDSAN	82.3	97.4	98.6	65.9	91.0	96.1	531.3	
Region+Bert	
Baseline	74.5	92.0	96.3	57.4	80.6	87.4	488.2	
CDSAN	84.1	97.7	99.1	69.7	93.0	97.0	540.6	

The implications of such research extend to numerous practical applications, with models exhibiting robust cross-dataset generalization capabilities being more likely to achieve successful implementation. These models demonstrate consistent efficacy across diverse data environments, mitigating the challenges posed by the variability of data characteristics and thereby accelerating the processes of deployment and optimization. For instance, integrating the CDGCN model into e-commerce recommendation systems allows for user-uploaded images and textual descriptions that may originate from disparate data distributions. By harnessing the insights garnered from cross-dataset generalization research, we can enhance and modify the model according to specific needs, improving its performance in recommendation tasks and enriching the user experience. Furthermore, the pursuit of advancements in cross-dataset generalization has catalyzed innovations in technology and methodology.

Visualization and qualitative result analysis

Visualization of visual and textual features: Utilizing a sample of 1,000 images from the Flicker30K test set, this work utilizes t-SNE (Van der Maaten & Hinton, 2008) to transform the acquired final visual features into a two-dimensional space. Following this, the K-means clustering technique is employed to group these features into 40 clusters. Each cluster is distinguished by a unique color, and the centroid of each cluster is annotated with its respective corresponding group number. The outcomes of visualization are illustrated in Fig. 6A, where this work highlights examples from certain clusters. As demonstrated in Fig. 6A, it is evident that images from different clusters exhibit distinct semantic signs, whereas those from the same cluster display consistent semantic symbols, such as young boys, adorable dogs, baseball and soccer, basketball sports, skiing, hats, all visually similar. For Group 20, featuring adorable dogs, and Group 37, focusing on skiing, there is a significant disparity in the two-dimensional feature map, indicating substantial differences in the features extracted by the proposed model for these two groups. Furthermore, despite Group 2, encompassing baseball and soccer, and Group 10, representing basketball and extreme sports, falling under the category of sports, they are still segregated into two classes due to minor distinctions, shown to be relatively close in the two-dimensional feature map. These visualizations underscore the proposed CDGCN model’s efficacy in effectively extracting global and distinctive local visual features, demonstrating strong global semantic coherence and highlighting its effectiveness.

Figure 6 (A) Visualization of learned visual-cross-modal representations via t-SNE on Flickr30K. (B) Two-dimensional visualizations of encoded cross-dimensional visual and textual representation vectors from the Flickr30K test set.

Detailed images with numbered divisions in (A) available at https://doi.org/10.5281/zenodo.14930469.

Concurrently, this work projects the 1,000 images and their corresponding top-ranking sentences into a shared two-dimensional space, as illustrated in Fig. 6B. The distribution of the dimensionality-reduced image vectors and text vectors appears largely similar. However, the visualization also reveals that a minority of image and text features are not mapped to the same distribution positions, reflecting potential matching errors during the model’s retrieval process.

Qualitative retrieval results visualization and analysis: This work conducted qualitative visualization and analysis of retrieval results on the Flickr30K dataset (detailed qualitative visualization and analysis figures available at https://doi.org/10.5281/zenodo.14930469). Due to the requirement of the journal that the retrieval visualization of the Flickr30K dataset must be accompanied by the permission signature of the person in the image, the author of the Flickr30k dataset is also unable to achieve it. Therefore, based on the Flickr30K dataset, the authors used their own portraits to create images (similar text descriptions) that were similar to the scene in the dataset, presenting the qualitative visualization of the model. CGDSA represents the Coarse-Grained Cross-Dimensional Semantic Dependency branch, FGDSD represents the Fine-Grained Cross-dimensional Semantic Aggregation branch, and CDGCN represents the Cross-Dimensional Coarse-Fine-Grained Complementary Network.

Qualitative retrieval results: Fig. 7 illustrates the top-5 text results retrieved by CGDSA, FGDSD, and CDGCN models for three query images, ranked by similarity scores. The proposed CDGCN model has retrieved almost all matching texts correctly (e.g., two out of three images matching perfectly, with only one text mismatch in the last image). Even in cases of incorrect matches, the semantic content of the text (e.g., words like “girls,” “lawn,” and “playing” in the incorrectly matched text for the third image) aligns well with the visual semantics of the image. In comparison, CGDSA and FGDSD models exhibit a higher incidence of mismatched texts. FGDSD method has incorrect matches in the second and three query images (e.g., failing to capture the image information “men and women” and incorrectly matching it with the text “several girls” in the second image). CGDSA method has mismatches in the first and third query images (e.g., not capturing the color of the hat and whether there are earrings in the first image, leading to incorrect matches with texts “brown and white hat” and “beige hat”). These findings indicate that the sentences retrieved by the CDGCN method provide detailed descriptions of visual scenes, suggesting that the image representations obtained by introduced model effectively capture rich dimensional feature information, avoiding mismatching issue by the complementary of coarse-fine-grained information.

Figure 7 The top 5 retrieval results of providing an image to retrieve sentences using the CGDSA, FGDSD, and CDGCN methods based on Flickr30k datasets using our own portrait pictures.

The retrieved five sentences are arranged in descending order of relevance. The sentences that match the given image are displayed in black font, while mismatched sentences are indicated in red font. Additionally, any missing matches are highlighted in green font. CGDSA represents the coarse-grained cross-dimensional semantic dependency branch, FGDSD represents the fine-grained cross-dimensional semantic aggregation branch, and CDGCN represents the cross-dimensional coarse-fine-grained complementary network.

As illustrated in Fig. 8, three sets of query texts are provided, each of which retrieves the top three images through CGDSA, FGDSD, and CDGCN models. These images are displayed from left to right by their degree of matching. The images enclosed in red boxes signify the correct retrievals derived from the given texts. This illustration demonstrates that the CDGCN model can largely retrieve the correct images (correctly retrieving two out of three texts, with one incorrect retrieval). For straightforward text-to-image retrieval tasks, all three models perform satisfactorily. However, for images that strongly emphasize the overall layout and comprehensive information (such as the first query text), both CGDSA and CDGCN adeptly match the correct images. In contrast, FGDSD overemphasizes minute details like “cake hat” and “candles are inserted into the cake,” thereby overlooking other contextual elements, leading to erroneous matches. In cases with extensive, intricate details (the second query text), CDGCN and FGDSD exhibit their strengths by accurately matching these details. Although CGDSA can also capture cross-dimensional feature information for broad-scale matching, it fails to achieve precise matching for certain detailed elements, resulting in failed matches. The analysis above underscores that the CDGCN method achieves a high matching rate in text-to-image retrieval, adeptly synthesizing local and global information to complement each other, thereby retrieving the ground truth images with greater accuracy.

Figure 8 The top 3 retrieval results obtained by inputting text for image retrieval using the CGDSA, FGDSD, and CDGCN methods based on Flickr30k datasets using our own portrait pictures.

The three retrieved images are arranged in descending order of relevance. The image that closely aligns with the provided text is accentuated within a red bounding box, whereas those that do not correspond are demarcated with black boxes. CGDSA represents the coarse-grained cross-dimensional semantic dependency branch, FGDSD represents the fine-grained cross-dimensional semantic aggregation branch, and CDGCN represents the cross-dimensional coarse-fine-grained complementary network.

Error analysis of retrieval results: As depicted in Fig. 7, the third given image resulted in errors across all three models: CGDSA, FGDSD, and CDGCN. Specifically, the text retrieved by the CDGCN method (Sentence 3) was mismatched, failing to align with the provided image. Despite this, the words “girls,” “lawn,” and “playing” within the sentence retained a high semantic relevance to the image content, carrying substantial semantic information closely related to the image. However, the actual content did not match due to discrepancies in minute details. The image showed three girls are lying on the lawn, while the retrieved text, though containing “girls,” “lawn,” and “playing,” omitted the crucial detail “setting.” For the missing correct text, while it included semantically relevant terms like “girls” and “lawn,” it failed to accurately capture verbs such as “huddled together” and “listening”, resulting in an inability to match the image. Similarly, as shown in Fig. 8 for text-to-image retrieval, given Text 3, all three models—CGDSA, FGDSD, and CDGCN—produced erroneous results. Specifically, the first two images retrieved by the CDGCN method are mismatched, while the third image is correctly matched. All three images were visually highly related, depicting a girl riding a bike, with different dressing details, movements, and categories of bikes. Figure 9 visualized examples of text-to-image retrieval errors, where words such as “girl,” “bike,” and “road” aligned well with corresponding visual regions in the images (though interpretations of “bike” varied). However, as seen in Fig. 9, third image (c), the verb “pedaling” lacked detailed semantic capture in the image, reducing the text-to-image match accuracy. Comparing the corresponding texts of the first two images (Text in black font in Fig. 9) with the query text revealed that the query sentence is relatively simple, encompassing basic semantic concepts found in all three images. The key verbs or actions in the first two images, “stepped,” “braking,” and “exploring,” “cruising,” are crucial semantic details missing from the query sentence, which the CDGCN failed to identify accurately.

Figure 9 Visualization of erroneous examples in text-to-image retrieval.

Words from the given sentence are highlightedaboutthe most relevant regions in the retrieved image, marked by red bounding boxes. The correct sentence that originally matched the retrieval image is indicated in black font, with crucial information differing from the given sentence encircled in yellow.

Based on the above analysis, the errors in retrieval results may stem from the CDGCN model’s issues in capturing verbs, actions, and anthropomorphized vocabulary in both images and texts, overlooking minor details that diminish matching accuracy. Enhancing feature extraction performance, strengthening the relationships between words and verbs or prepositions to capture more local features, and improving the model’s architecture to enhance global contextual understanding (including anthropomorphized sentences) could potentially elevate the accuracy of image-text retrieval.

Conclusion

This work proposes an image-text matching method called Cross-Dimensional Coarse-Fine-Grained Complementary Network. The framework integrates a coarse-grained overall cross-dimensional semantic aggregation branch with a fine-grained local cross-dimensional semantic dependency branch to form establish a holistic approach that ensures consistency between global and local semantic information. A plug-and-play coarse-grained cross-dimensional semantic aggregation strategy is also developed to deeply analyze the importance and relationship between different dimensions of the same feature and various features in the fixed dimensional attention space. The proposed CDGCN has been validated on two benchmark datasets to extract richer features from image text, avoiding the problem of mismatch between overall and local features, and enhancing the efficacy of image-text matching techniques. Despite the progress achieved by the proposed model, certain limitations remain apparent. Firstly, our reliance on traditional methodologies for model construction may not constitute the most optimal solution. In the future, the integration of currently popular pre-trained vision-language large models to establish a “multi-modal model fusion network” could potentially extract more representative image and text features, enhancing retrieval accuracy and efficiency. Furthermore, our current focus exclusively on cross-modal image-text matching presents an opportunity to expand towards multi-modal retrieval, optimizing the CDGCN model to undertake cross-modal video-text retrieval tasks. In addition, based on error analysis, the CDGCN model will be optimized and planned to be applied to an intelligent search system embedded in smart education for actual campus projects.

Supplemental Information

Supplemental Information 1 Code.

Additional Information and Declarations

Competing Interests

The authors declare that they have no competing interests.

Author Contributions

Meizhen Liu conceived and designed the experiments, performed the experiments, analyzed the data, performed the computation work, prepared figures and/or tables, authored or reviewed drafts of the article, and approved the final draft.

Anis Salwa Mohd Khairuddin conceived and designed the experiments, authored or reviewed drafts of the article, and approved the final draft.

Khairunnisa Hasikin performed the computation work, authored or reviewed drafts of the article, and approved the final draft.

Weitong Liu performed the computation work, authored or reviewed drafts of the article, and approved the final draft.

Data Availability

The following information was supplied regarding data availability:

Access to the Flickr30k dataset can be requested at https://forms.illinois.edu/sec/229675.

The MSCOCO dataset is available at Microsoft COCO: Common Objects in Context: https://cocodataset.org/#download.

Code is available at https://github.com/meizhenliu/CDGCN-code and Zenodo AAAAAkunkun. (2025). meizhenliu/CDGCN-code: CDGCN-code (qmD6CkPYK.ZkVe9). Zenodo. https://doi.org/10.5281/zenodo.14930469.

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
