# Peer review of "Novel cross-dimensional coarse-fine-grained complementary network for image-text matching"

_PeerJ Computer Science, doi:10.7717/peerj-cs.2725_

## Round 0.1 · original submission · Major Revisions

Please review the suggestions made by the reviewers and take appropriate action. Please revise the paper, submit a paper with changes highlighted, and a response to each significant suggestion. I will have a second review done before making a final decision. Thanks for your interest in the journal.

Reviewer 1 ·

Basic reporting

1. The manuscript requires extensive formatting, like justified, cross-referencing, font size (Check for line 25 in the abstract: font size seems smaller than the rest of the section), etc.
2. No future scope is mentioned.
3. The plagiarism of the manuscript is more than 15%.
4. The manuscript must be checked via some Grammarly tool.
5. Authors must include the challenges, significance, and motivation section.
6. The contribution points must be short and meaningful.

Experimental design

1. The hyperparameters are reported but how are they defined?
2. An error analysis could be done to add more meaning to the proposed architecture. Moreover, any qualitative visualization can also be done.
3. Provide more context on the significance and impact of cross-dataset generalization study findings.
4. Expand on the practical implications of the study's superior outcomes for real-world applications.
5. Provide the data and program to validate the reported results via GitHub.
6. Some relevant research can be added to strengthen the literature such as Multimodal hate speech detection via multi-scale visual kernels and knowledge distillation architecture, AW-MSA: Adaptively weighted multi-scale attentional features for DeepFake detection.

Validity of the findings

No Comment

Additional comments

No Additional Comments

Reviewer 2 ·

Basic reporting

This work learns the image-text matching. They design a new network, cross-dimensional coarse-fine-grained complementary network (CDGCN), to combine information from the image and the text modalities, ensuring good performances in retrieval tasks. The structure of the paper is easy to understand, and several related works are discussed, cited and compared with. The code is also provided.

However, there are still some improvements that can be made. First, the motivation should be explained more. For example, the paper mentions in the introduction "This pooling methods ignore the fact that different dimensions of a feature could carry distinct interpretations, and the significance of various dimensions with the same feature might differ, which may further reduce discriminative information.", but no evidence or related work is attached for this assumption. Then, the writing should be polished as well, and some typos still exist in the submitted manuscript. For example, in line 301, the "m salient image regions {v_p}^{n}_{p=1}" should be "m salient image regions {v_p}^{m}_{p=1}". In the Figure 6's caption, "he retrieval results of providing ..." should be "The retrieval results of providing ...". I also suggest the authors to revisit all the figures' captions since right now details are missed in them. For example, in table 1, even in the main paper the evaluation metrics are explained, it will be nice to emphasize them in the captions as well. I also feel confused about the term "ensemble*" in the table 1-5, and hope the authors can provide more details of this experiment. Finally, for the literature review, it will be more interesting if the paper also discusses several pre-trained VLMs that can also do retrieval tasks, such as BLIP2. Even though these works may not compare with the proposed method directly.

Experimental design

The code is provided to reproduce the results. No comment.

Validity of the findings

Open-sourced datasets are used. No comment.

---

## Round 0.2 · accepted · Accept

I am conditionally accepting your paper. Please incorporate the appropriate available results for the suggestion made by the reviewer while in production.

Reviewer 1 ·

Basic reporting

The manuscript can be accepted with no further changes.

Experimental design

no comment

Validity of the findings

no comment

Reviewer 2 ·

Basic reporting

no comment

Experimental design

no comment

Validity of the findings

no comment

Additional comments

Thank the authors for providing with the discussion about VLMs such as BLIP-2. Even though I think it will be more interesting if the authors can add the text-to-image and image-to-text retrieval results in the results' tables such as Table 1 since these VLMs also have (zero-shot) experimental results for benchmarks such as Flicker30k. For example, BLIP-2's zero-shot performances on Flickr30k-test set is 97.6 (TR@1) and 89.7 (IR@1). In Table 1, it is hard for me to agree the shown baselines belong to "state-of-the-art methods" since none of them gets better results than BLIP-2 zero-shot results.